# Hypertension, Diabetes and Depression as Modifiable Risk Factors for Dementia: A Common Data Model Approach in a Population-Based Cohort, with Study Protocol and Preliminary Results

**DOI:** 10.3390/jcm14186622

**Published:** 2025-09-19

**Authors:** Corrado Zenesini, Silvia Cascini, Roberta Picariello, Francesco Profili, Laura Maria Beatrice Belotti, Laura Maniscalco, Anna Acampora, Roberto Gnavi, Paolo Francesconi, Luca Vignatelli, Francesco Nonino, Annamaria Bargagli, Domenico Tarantino, Giuseppe Salemi, Nicola Vanacore, Domenica Matranga

**Affiliations:** 1IRCCS Istituto delle Scienze Neurologiche di Bologna, 40139 Bologna, Italy; lauramariabeatrice.belotti@ior.it (L.M.B.B.); l.vignatelli@ausl.bologna.it (L.V.); f.nonino@ausl.bologna.it (F.N.); 2Department of Epidemiology, Lazio Regional Health Service, 00147 Rome, Italy; s.cascini@deplazio.it (S.C.); a.acampora@deplazio.it (A.A.); a.bargagli@deplazio.it (A.B.); 3Servizio Sovrazonale di Epidemiologia ASL TO3, Regione Piemonte, Collegno, 10093 Turin, Italy; roberta.picariello@epi.piemonte.it (R.P.); roberto.gnavi@epi.piemonte.it (R.G.); 4Agenzia Regionale di Sanità Della Toscana, 50141 Florence, Italy; francesco.profili@ars.toscana.it (F.P.); paolo.francesconi@ars.toscana.it (P.F.); 5Department of Health Promotion, Mother and Child Care, Internal Medicine and Medical Specialties, University of Palermo, 90127 Palermo, Italy; laura.maniscalco04@unipa.it (L.M.); domenica.matranga@unipa.it (D.M.); 6Dipartimento di Radiologia Diagnostica, Interventistica e Stroke, Azienda Ospedaliera Universitaria, Policlinico, 90127 Palermo, Italygiuseppe.salemi@policlinico.pa.it (G.S.); 7National Center for Disease Prevention and Health Promotion, National Institute of Health, 00161 Rome, Italy; nicola.vanacore@iss.it

**Keywords:** dementia, common data model, population-based cohort

## Abstract

**Background/Objectives**: Dementia is a major public health challenge, with age as its primary non-modifiable risk factor. Several modifiable conditions, such as hypertension, diabetes, and depression, have been identified as potential targets for prevention. The aim is to describe the methodology and preliminary results of a study that will be conducted within the Italian National Health Service (INHS), designed to assess the impact of hypertension, diabetes, depression, and their interactions on the onset of dementia. **Methods**: This population-based cohort study, part of the PREV-ITA-DEM project, was conducted using a Common Data Model (CDM) approach across five Italian regions and cities participating in the NeuroEpiNet network. Individuals aged ≥ 50 years without prior diagnoses of dementia, depression, diabetes, or hypertension were followed from cohort entry (2011–2013) until dementia diagnosis, death, emigration, or study end (2019–2022). Exposures were time-dependent and defined using validated algorithms applied to Healthcare Utilization Databases (HUDs). Associations between chronic conditions and dementia risk will be estimated using competing risks regression models adjusted for confounders. **Results**: The final cohort comprised more than 3 million individuals, with a mean baseline age of 63–65 years and a female proportion of 52–55%. On 1 January 2011, the prevalence of individuals aged ≥ 50 years with dementia ranged from 8.7 to 14.7 per 1000 population. A harmonized methodological framework based on a CDM was developed and implemented across all sites, incorporating a shared protocol, standardized local databases, and uniform analytic scripts, and the results will be pooled using meta-analytic techniques. **Conclusions**: Preliminary findings confirm the feasibility of a standardized, multi-regional CDM approach and the potential for HUDs to support large-scale dementia prevention studies in real-world settings.

## 1. Introduction

Dementia is a neurodegenerative disease characterized by progressive cognitive and functional decline, ultimately leading to loss of independence and the need for continuous care, frequent hospitalizations, and increased mortality [1]. It results from diverse pathological changes in the brain, including the accumulation of abnormal proteins, neuronal loss and vascular damage. The most common types of dementia include Alzheimer’s disease, characterized by amyloid-beta plaques and neurofibrillary tau tangles; vascular dementia, caused by cerebrovascular disease; Lewy body dementia, marked by alpha-synuclein-containing Lewy bodies; and frontotemporal dementia, involving degeneration of the frontal and temporal lobes [2]. Preventing dementia remains challenging, as its causes are often not fully understood. The primary non-modifiable risk factor of dementia is age, which, combined with global population aging, is expected to drive a significant increase in dementia cases worldwide [3].

Numerous studies highlight that certain modifiable risk factors could delay or even prevent the onset of dementia [4,5]. In 2017, two major reports, from the Lancet International Commission and the Agency for Healthcare Research and Quality, identified several potentially modifiable risk factors for dementia [6,7,8]. The strength of evidence varied among factors, but it was robust for many of them. The Lancet Commission identified nine modifiable risk factors acting across different stages of life: low educational attainment, hypertension, untreated hearing loss, obesity, smoking, depression, physical inactivity, diabetes, and low social engagement. These reports emphasize that addressing such risk factors, particularly during midlife (ages 45–65), could potentially delay or prevent the onset of dementia, with an estimated 35% of global cases being preventable.

In the Italian context, the combined Population Attributable Fraction (PAF) for 11 modifiable risk factors was estimated at 39.6% (95% CI: 20.8–55.9); a 10% reduction in these risk factors would prevent 54,495 dementia cases, with a Potential Impact Fraction (PIF) ranging from 3.7% to 6.0% [9]. An updated 2024 report from the Lancet Commission expanded the list to 14 risk factors, adding alcohol consumption, air pollution, and traumatic brain injury [10]. While additional evidence is still required, these findings support the potential of promoting a healthy lifestyle in the prevention of dementia.

In recent decades, population-based studies strengthened the role of primary prevention by reporting a decrease in the incidence of dementia, potentially attributable to reduced exposure to certain risk factors [11,12,13,14]. The 12 risk factors identified by the Lancet Commission were supported by the best available international scientific evidence, although considerable heterogeneity persisted across studies. Therefore, population-based research remains essential to assess the impact of putative risk factors on dementia onset and to evaluate their preventive relevance within specific contexts.

In recent years, several predictive models were developed and validated in population-based studies to examine the relationship between risk factors and the onset of dementia. Most of them [15,16,17] integrated modifiable and non-modifiable risk factors, including specific biomarkers, showing variable degrees of predictive accuracy (C-statistics: 0.65–0.86), with variability possibly reflecting substantial heterogeneity in study populations, together with limitations related to implementation, such as the need for ad hoc data collection. In Italy, such data are not routinely available through active national surveillance or standard healthcare databases.

These findings underscore the major challenges faced by the scientific and public health communities in addressing the needs of current and future dementia patients, especially in the area of primary prevention [3,18].

In response to calls by researchers and international health authorities for greater attention to neurological patients during the COVID-19 pandemic, a multidisciplinary, interregional, and inter-institutional initiative, *NeuroEpiNet* (Clinical Epidemiology Network for Chronic Neurological Diseases), was established in Italy with the aim to generate robust evidence form the available Healthcare Utilization Databases (HUDs). The network currently includes eight regions (Piedmont, Veneto, Emilia-Romagna, Tuscany, Umbria, Lazio, Apulia and Sicily) and brings together neurologists, epidemiologists, pharmacoepidemiologists and biostatisticians from Italian National Health Service (INHS) institutions, research centers and academic partners [19]. The NeuroEpiNet methodological approach was based on a shared Common Data Model (CDM), incorporating standardized protocols, harmonized HUDs, and the use of a unified data processing script implemented locally at each participating center. CDMs are frameworks designed to standardize terminologies related to medication use, medical events, procedures and data structures, thereby facilitating consistent analyses across multiple data sources and databases. CDMs are characterized by three key features: (i) adaptability to specific research questions; (ii) transparency to enable reproducibility, validity assessment and stakeholder confidence; and (iii) usability to support efficient and timely analyses [20].

The aim is to describe the methodology and preliminary results of a population-based cohort study that will be conducted within the Italian healthcare system, designed to assess the impact of hypertension, diabetes, depression, and their interactions on the onset of dementia, using a CDM approach.

## 2. Materials and Methods

The study protocol was approved by the National Ethical Committee (AOO-ISS 0017087) on 6 April 2023 and was part of the *PREV-ITA-DEM project*, a large-scale study funded in 2022 by the Italian Ministry of Health under the National Recovery and Resilience Plan (PNRR-MAD-2022-12375822), partnered by *NeuroEpiNet network*.

The STROBE (Strengthening the Reporting of Observational Studies in Epidemiology) [21] and the RECORD (The REporting of studies Conducted using Observational Routinely collected health Data) guidelines [22] were followed.

### 2.1. Study Design

This was a multicenter, historical cohort study based on a record linkage approach between HUDs, using a CDM approach.

### 2.2. Population and Setting

The institutional members of the *NeuroEpiNet network* [19] include the IRCCS Institute of Neurological Sciences of Bologna (Emilia-Romagna), the Department of Epidemiology of the Latium Regional Health Service, the Epidemiology Unit of Turin Local Health Trust (LHT) TO3 (Piedmont), the Epidemiology Unit of Tuscany Regional Health Agency, and the Epidemiology Unit of Trapani LHT (Sicily). The cohort included all individuals aged ≥ 50 years who were residents and eligible for healthcare coverage in the participating areas. The starting date of cohort entry varied by region according to the availability of administrative data: 1 January 2011 for Bologna, Tuscany and Latium, and 1 January 2013 for Piedmont and Trapani. Participants were followed from cohort entry until the earliest occurrence of one of the following outcomes: dementia onset, death, out-migration, or the end of the study period (31 December 2022 for Bologna, Latium and Tuscany and 31 December 2019 for Piedmont and Trapani). The maximum follow-up duration was 12 years.

### 2.3. Exposure and Outcome

At baseline, all participants were considered unexposed to the conditions of interest and dementia-free. During follow-up, individuals were classified as “exposed” once they met some algorithm-defined criteria for depression [23], diabetes [24] and hypertension [25], based on HUDs established by the *NeuroEpiNet network*. From the date of the first exposure, individuals were classified as exposed and remained in this status for the duration of follow-up period. Each participant contributed person-time according to his or her time-varying exposure status. In addition to single-exposure analyses, multiple exposures were examined. In the Latium region, an additional exposure related to the COVID-19 pandemic was assessed. For this analysis, a sub-cohort was constructed following the same inclusion and exclusion criteria but with follow-up beginning on 1 January 2020 and ending on 31 December 2022. The outcome was incident dementia, identified using a validated algorithm [26]. A four-year look-back period (1 January 2007 to 31 December 2010) was applied to exclude prevalent dementia, hypertension, diabetes and depression cases and ensure that only incident cases were identified during follow-up (Figure 1).

### 2.4. Data Sources and Common Data Model Approach

All data were derived from HUDs managed within the INHS. To ensure compliance with the European Union General Data Protection Regulation (GDPR), Italian privacy laws, and the specific provisions outlined in *Garante Privacy, Provvedimento n. 298/2024* (*https://www.garanteprivacy.it/home/docweb/-/docweb-display/docweb/10016146 accessed on 10 August 2025*), all personal identifiers were removed or pseudonymized prior to data analysis, preventing individual subject identification. Data access was strictly limited to authorized research personnel, and data processing adhered to principles of confidentiality, data minimization and security (GDPR-compliant server). To identify the population, the exposure cohorts and incident dementia cases, the following administrative health data sources were utilized:Hospital Discharge—containing primary and secondary diagnoses coded using the International Classification of Diseases, Ninth Revision, Clinical Modification (ICD9CM).Pharmaceutical Prescription—including both community and direct hospital pharmacy dispensing of all medications reimbursed by the INHS coded using Anatomic Therapeutic Chemical (ATC) codes for drug classification; the ATC system is the drug classification system adopted by the World Health Organization [27].Co-Payment Exemption—listing individuals certified by an INHS specialist as having a disease qualifying for medical co-payment exemption.Demographic Population Registry—identifying all residents under the INHS, including death and out-migration information.

Data were linked using a deterministic record linkage approach based on a unique, anonymized identifier generated through local pseudo-anonymization procedures.

A harmonized methodological framework based on a CDM was implemented across all study sites (Figure 2). This approach included the use of a shared, pre-specified protocol, the development of standardized local databases and consistent data analysis using a shared script. Each center will process its data independently and generate aggregated, anonymized results, which will subsequently be pooled using meta-analytic techniques. All procedures were carried out in accordance with national and regional data protection regulations.

### 2.5. Statistical Analysis

A competing risks regression model will be used to assess the association between exposure categories and the risk of incident dementia, accounting for death as a competing event. Exposure will be defined as a time-dependent categorical variable representing all possible combinations of three chronic conditions: depression, diabetes, and hypertension. Individuals will be categorized into eight mutually exclusive groups based on the presence or absence of each condition during follow-up: no exposure, depression only, diabetes only, hypertension only, depression and diabetes, depression and hypertension, diabetes and hypertension and all three conditions (Figure 3). Age will be recalculated dynamically as a time-varying covariate based on baseline age at each risk interval.

Data will be analyzed using Fine and Gray sub-distribution hazard models [28,29], with dementia as the event of interest and death as the competing event. The model will adjust for age, sex, geographic area (district or province of residence) and comorbidities measured using the Charlson Comorbidity Index, which was calculated based on hospital discharge data [30]. The results will be presented with Sub Hazard Ratio (SHR) and 95% Confidence Interval (95% CI). The analyses will be stratified by sex.

A sensitivity analysis will be conducted with a 3-year lag between exposure and outcome to minimize potential reverse causality [31].

Statistical analyses will be performed using Stata/SE 14.2 (StataCorp LP, College Station, TX, USA) software and SAS 9.4, Service Pack 8 (SAS Institute Inc., Cary, NC, USA) software.

### 2.6. Meta-Analysis

Separate analyses will be conducted for each participating city or region. Local results will then be aggregated to estimate the overall risk across cohorts using a meta-analytic approach. Specifically, the inverse-variance weighted method will be used to combine the local SHR estimates, adjusting for the precision of each study. A random-effects model will be applied to account for variability between areas, and heterogeneity will be assessed using the I^2^ statistic. The overall pooled SHR and 95% Confidence Interval will be computed from the meta-analysis of local estimates, providing a comprehensive estimate of the association between exposure categories and dementia risk across all cohorts.

## 3. Results

As of 1 January 2011, the study included 372,105 individuals aged ≥ 50 residing in the Local Health Trust of Bologna, 2,235,351 in the Latium region and 1,666,431 in the Tuscany region. As of 1 January 2013, the study included 2,012,056 individuals aged ≥ 50 residing in the Piedmont region and 181,102 in Trapani. The starting year of follow-up varied across centers, depending on the availability of health administrative databases. After applying a look-back period, excluding prevalent people with depression, hypertension, diabetes and dementia, the final cohort included 181,998 subjects in the Local Health Trust of Bologna, 1,056,717 in the Latium region, 850,286 in the Tuscany region, 927,725 in the Piedmont region and 43,882 in the LHT of Trapani (Figure 4) (overall 3,152,368 included subjects). On 1 January 2011, the prevalence of individuals aged ≥ 50 years with dementia ranged from 8.7 to 14.7 per 1000 population.

Demographic characteristics were comparable across cohorts, with a mean age ranging from 63.1 to 64.8 years and the proportion of female participants ranging from 52.1% to 54.8% (Table 1).

### 3.1. Common Data Model

In each Italian health region, administrative databases are managed through different informatics platforms. Accordingly, each participating area relied on independently developed systems with differing structures, coding schemes and data formats. To enable consistent multi-regional analyses, a comprehensive harmonization process was undertaken to integrate these heterogeneous sources into a unified CDM. This process required aligning administrative health data across five core domains, linked by a patient identification code: hospital discharge, drug prescriptions, exemption registries, demographic information and the mortality registry. For the hospital discharges, the mandatory variables were as follows: principal and secondary diagnosis codified by ICD-9 codes, main and secondary operation/procedures codified by ICD-9 codes, dates of admission and discharges, type of discharge (discharge, transferred, deceased and other) and admission regime (ordinary, day-hospital, home treatment, day-surgery with overnight stay); for the drug prescriptions, drug dispensing date, Anatomical Therapeutic Chemical (ATC) classification code, drug code, prescription coverage days, number of packages and mode of dispensing (territorial or direct); for the exemption for pathology, the exemption code from the National Institute of Statistics (ISTAT), start and end dates; for the health demographic database, sex, date of birth, residence start and end date, region and municipality of residence (both with ISTAT code) and date of death (Table 2).

### 3.2. Algorithm Definition

For each pathology, the onset was defined as the earliest date on which one of the identification criteria was met in the HUDs.

#### 3.2.1. Dementia

The *NeuroEpiNet* network chose a validated algorithm [26] to identify people with dementia. The algorithm classifies subjects as having dementia if they meet at least one of the following criteria: at least two different prescriptions of drugs for dementia within 12 months or at least one hospital discharge with primary or secondary diagnoses of dementia or mild cognitive impairment or if the subjects had the exemption from healthcare co-payment specific for the disease (Table 3). It was validated on a population of over 1100 clinically diagnosed dementia and mild cognitive impairment cases and a matched control group across four Italian regions, showing a sensitivity of 74.5% and specificity of 96.0%.

#### 3.2.2. Depression

As no validated algorithm for identifying depression based on HUDs was available within the Italian health system, the *NeuroEpinet* group adopted an algorithm used in previously published studies [23]. According to this algorithm, individuals were classified as having depression if they had either at least 180 days of antidepressant use within a 12-month period or hospital discharge diagnosis of depression based on specific ICD-9 codes [32] (Table 3).

#### 3.2.3. Diabetes

The *NeuroEpiNet* network employed an algorithm originally developed by Gnavi et al. [24] and recently validated [33] to identify individuals with diabetes. According to this algorithm, a subject was classified as having diabetes if they met at least one of the following criteria: at least two prescriptions of antidiabetic medications within a 12-month period, or at least one hospital discharge with a primary or secondary diagnosis of diabetes, or exemption from healthcare co-payment specific to diabetes (Table 3). The algorithm was validated in a sample of 1545 individuals from various areas of the Latium region, showing a sensitivity of 90.9% and a specificity of 97.4% [33].

#### 3.2.4. Hypertension

The *NeuroEpiNet* network adopted a validated algorithm developed by the Tuscany Regional Health Agency [25], which is incorporated into the MaCro data warehouse on chronic diseases. According to this algorithm, individuals were classified as having hypertension if they met at least one of the following criteria: at least two prescriptions for antihypertensive medications within a 12-month period, at least one hospital discharge with a primary or secondary diagnosis of hypertension, or exemption from healthcare co-payment specific to hypertension (Table 3). To increase the specificity of the algorithm, individuals with a diagnosis of heart failure were excluded.

## 4. Discussion

This study describes the methodology and preliminary results of a large, population-based investigation conducted within the Italian healthcare system, aimed at evaluating the impact of hypertension, diabetes, depression, and their combination on the onset of dementia. Three regions and two cities, from the north to the south of Italy, were included in the study for 3,152,368 people.

In this cohort study, a competing risks regression model will provide a robust framework to assess the relationship between exposure factors and the risk of incident dementia, while accounting for death as a competing event. By modeling exposure as a time-dependent categorical variable, the analysis will capture dynamic changes in individuals’ chronic disease status over time, thereby more accurately reflecting real-world disease progression. The Fine and Gray sub-distribution hazard model [28,29] will allow for appropriate estimation of dementia risk in the presence of competing mortality, a consideration particularly relevant in older populations. Age will be adjusted for as a time-varying covariate to enhance the precision of risk estimates by reflecting the changing baseline hazard over the course of follow-up. Stratification by sex and adjustment for key confounders, including comorbidity burden and geographic region, will help reduce potential bias. A sensitivity analysis incorporating a 3-year lag between exposure and outcome will be conducted to strengthen causal inference by mitigating the risk of reverse causality. The 3-year lag was selected based on the clinical history of dementia, recognizing that the prodromal phase, which includes subtle cognitive decline and other early symptoms, typically spans several years before a formal diagnosis is established [34]. A systematic review and meta-analysis [31] reported an average time to diagnosis of 3.5 years (95% Confidence Interval: 2.7–4.3 years) from symptom onset. This interval reflects the delay between initial symptom manifestation and clinical recognition.

During the design of the study and the analysis plan, several context-specific factors needed to be taken into consideration. Firstly, we had to consider the substantial procedural heterogeneity of HUD management across regions in the Italian context, due to the relative administrative autonomy of each Italian region, allowing for the adoption of different systems and informatics infrastructures for managing and utilizing administrative health databases. To address this potential barrier, the *NeuroEpiNet* network adopted a CDM approach that provides a standardized data structure, harmonizing the underlying logical infrastructure enables consistent information exchange across diverse centers and data sources and allows analytical tools and statistical scripts to operate within a unified framework. In order to allow for the exchange of analytical methodologies rather than individual-level data, after data harmonization, each research group independently analyzed its local dataset by applying a shared and validated statistical script to a structured database. Such an approach allowed for compliance with data privacy and methodological consistency. At the same time, the CDM framework enhanced transparency and reproducibility and will allow other research groups to replicate the study in their own populations by aligning their datasets with the same structure and implementing the shared script [20].

To our knowledge, no other Italian networks currently apply the CDM methodology to health administrative databases specifically for the study of neurological diseases. However, several national initiatives in other clinical domains have successfully implemented CDM-based approaches, demonstrating the feasibility and utility of standardized data integration across heterogeneous healthcare systems. In this context, the Italian Medicines Agency (AIFA) has recently promoted the creation of a network, including eight Italian regions and several experts from both public and academic Italian institutions, called MoM-Net (Monitoring Medication Use During Pregnancy Network), focusing on monitoring medication use in pregnancy, through the integration of different regional health databases [35]. Another example is the VALORE project, a large-scale multi-database network that provides access to data on more than 140,000 biological drug users with immune-mediated inflammatory diseases from 13 Italian regions [36].

Another challenge the *NeuroEpiNet* network had to face was the substantial heterogeneity in the Italian setting regarding the use of algorithms on administrative health databases to identify people with pathologies. Systematic reviews of the literature have highlighted differences among algorithms developed in Italy for conditions such as diabetes [37] and hypertension [38], whereas no conclusive evidence was identified for algorithms related to dementia or depression. One of the key steps performed by the *NeuroEpiNet* network was selecting the most appropriate case identification algorithms for each condition by means of a multidisciplinary approach involving neurologists, methodologists, pharmacologists and statisticians. For dementia and diabetes, the group selected validated algorithms that had been tested vs. clinical diagnoses and showed good diagnostic accuracy [26,33]. No validated algorithms were identified for depression; therefore, the group adopted a previously used approach developed by the Bologna research team, which defines cases based on a threshold of at least six months of antidepressant use within a year [23]. For hypertension, a validated algorithm developed by the Tuscany Regional Health Agency was adopted, which is incorporated into the MaCro data warehouse on chronic diseases [25].

A key strength of this study is the use of multiple health administrative databases to identify cohorts affected by a specific condition, enabling access to large, population-based datasets with the capacity for long-term follow-up. When appropriately managed and harmonized, administrative data are cost-effective, readily accessible, and can provide real-world insights across different healthcare settings within significantly shorter timeframes compared to ad hoc clinical studies [39]. This advantage may be particularly valuable when evidence is rapidly needed to inform healthcare policies in a timely manner, such as during public health emergencies. In this perspective, our findings may be important for decision makers while recommending guidance on lifestyle changes and aggressive treatment of conditions that may be avoidable risk factors for preventing dementia. One other potential advantage of a population-based study on administrative data like ours is the possibility of adapting its methods to other health systems with similar characteristics. Italy has a universal, single-payer healthcare system, where, unlike insurance-based ones, access to healthcare and the mode of its delivery are relatively standardized, favoring homogeneity in the stage of data retrieval. INHS is organized into three levels: national; regional (21 regions); and local (on average, 10 local health units per region). Administrative data on healthcare reimbursed by the system, such as inpatient care and drug dispensations, are routinely collected by local health units and, in some regions, sent to the regional level [40].

Our study has several limitations. Firstly, there is potential for misclassification or underreporting of diagnoses, as coding practices may vary and are often influenced by administrative or billing priorities rather than clinical accuracy. Specifically, misclassification of depression may have occurred because antidepressant use was determined based on prescription records, which do not confirm actual medication intake or adherence to prescribed dosages. However, we defined depression exposure as having at least six months of continuous prescriptions, making it unlikely that individuals with such extended prescriptions did not take the medication. Additionally, while antidepressants can be prescribed for conditions other than depression, these alternative indications are generally treated with shorter courses, reducing the likelihood that long-term prescriptions represent non-depressive conditions. Similarly, for other conditions where validated algorithms were applied, some degree of misclassification may have occurred. To minimize this limitation, we selected algorithms with high specificity. Such potential misclassification is likely non-differential, which would tend to bias risk estimates toward the null. We will further assess the impact of these potential misclassifications during the analysis of association estimates. Secondly, the absence of detailed clinical information, such as symptom severity or diagnostic test results, may reduce the specificity and sensitivity of case identification. Therefore, while administrative data are valuable for epidemiologic research, validation against clinical records or disease registries is often necessary to ensure diagnostic accuracy. Nonetheless, information on other potential confounders, such as lifestyle factors (e.g., smoking and physical activity), socioeconomic status and education [10], was not available in the administrative data; therefore, our estimates of risk factor effects on the outcome may be subject to residual confounding. Lastly, one additional challenge may arise from the heterogeneity in informatic platforms and procedures that may be encountered at a country level. The need for harmonization of such technical aspects may limit the feasibility and timeliness of projects addressing health issues by means of administrative databases in population-based settings.

Despite these limitations, our analytical approach offers valuable insights into the cumulative impact of common chronic conditions on dementia risk within a large, real-world population. Its findings may provide a valuable contribution to the knowledge of hypertension, diabetes and depression as risk factors for dementia, informing public health guidance on how to mitigate them.

## Figures and Tables

**Figure 1 jcm-14-06622-f001:**
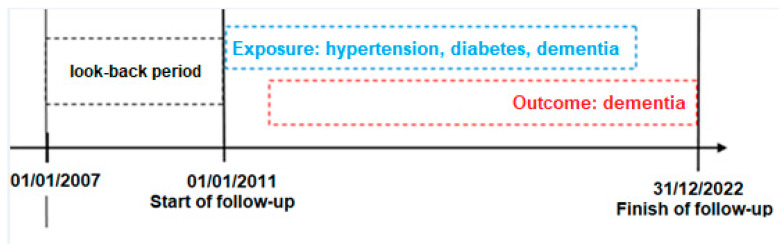
Study design. The dashed boxes represent the look-back period, exposure and outcome timeframes.

**Figure 2 jcm-14-06622-f002:**
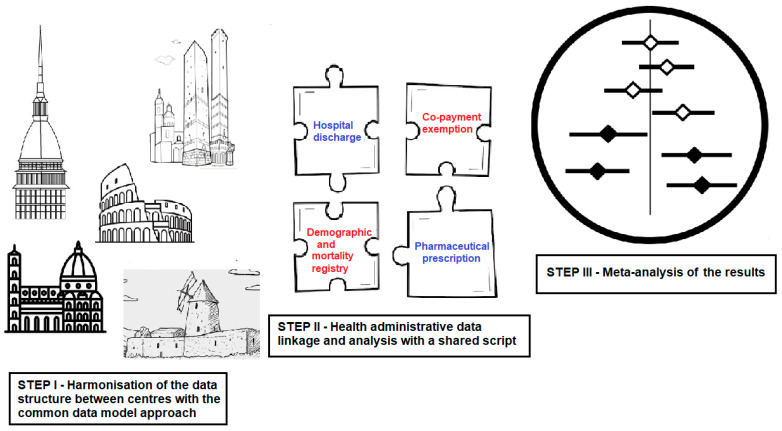
Methodological and analytical approach used within the *NeuroEpiNet* network.

**Figure 3 jcm-14-06622-f003:**
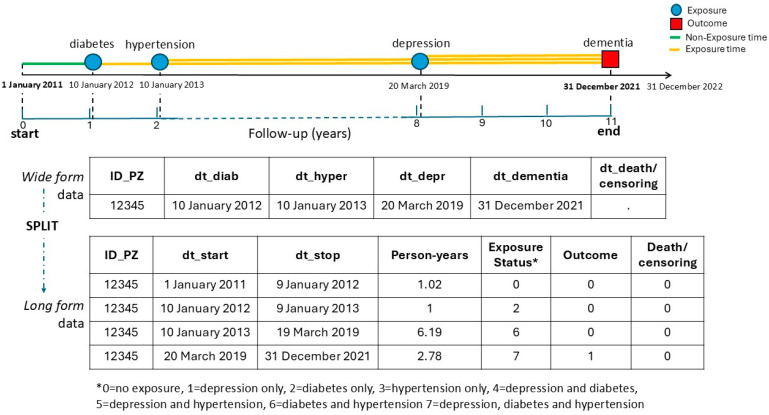
Time-dependent variables with multiple exposure. The dashed arrow represents the sequence of data transformation steps: from wide to long format. The solid arrow represents the progression of time.

**Figure 4 jcm-14-06622-f004:**
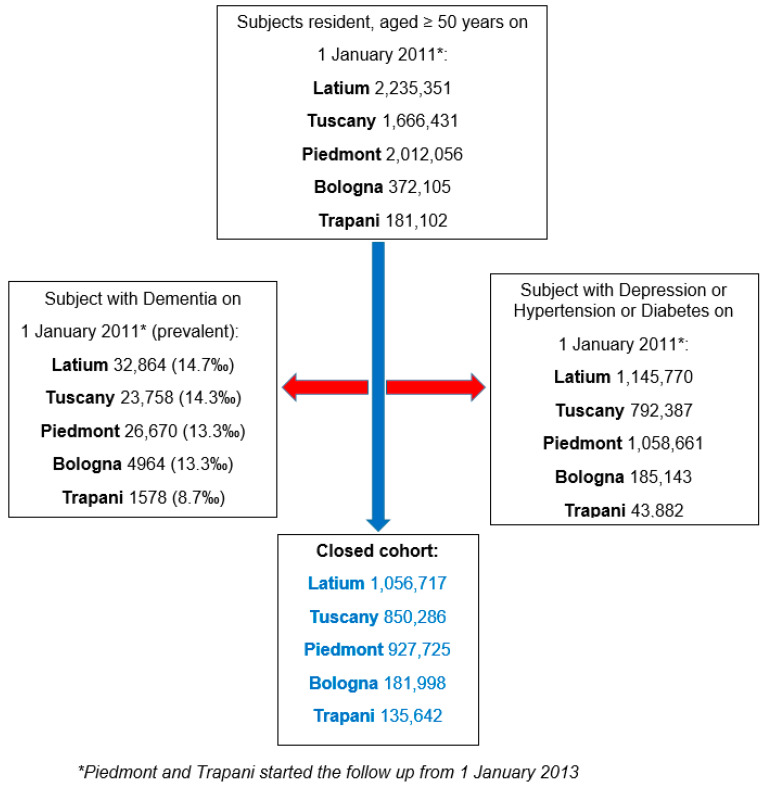
Cohort recruitment flowchart. Arrows indicate the flow and sequence of steps in the recruitment process.

**Table 1 jcm-14-06622-t001:** Demographic characteristics of the study cohort in the five populations.

Regions	Cohorts Population-N	Age-Years Mean (SD)	Sex: Female *n* (%)
Piedmont	927,725	63.6 (11.4)	483,182 (52.1)
Latium	1,056,717	63.1 (10.9)	572,024 (54.1)
Tuscany	850,286	64.8 (11.4)	456,465 (53.7)
Bologna (Emilia-Romagna)	181,998	63.1 (10.4)	99,786 (54.8)
Trapani (Sicily)	135,642	65.2 (11.2)	72,990 (53.8)

**Table 2 jcm-14-06622-t002:** Common Data Model encoding scheme for hospital discharge, drug prescriptions, exemption for pathology, health demographic population registry and mortality registry in the Healthcare Utilization Databases.

Linkage Key for All Health Administrative Databases			
Variable Name	Definition	Variable Type (Length)	Encoding
*codice*	*Patient identification code-linkage key*	*Character*	*Alphanumeric*
**Hospital Discharge (Schede di Dimissioni Ospedaliera)**			
diag	Principal diagnosis	Character (5)	ICD-9-CM
diagsec1-5	From 1st to 5th secondary diagnosis	Character (5)	ICD-9-CM
interv	Main surgical intervention	Character (4)	ICD-9-CM
interv1-5	From 1st to 5th secondary operation/procedure	Character (4)	ICD-9-CM
data_amm	Date of admission	Date (10)	dd/mm/yyyy
data_dim	Date of discharge	Date (10)	dd/mm/yyyy
tip_dim			0: discharge
	Type of discharge	Character (1)	1: transferred
			2: deceased
			3: other
regric			1: ordinary
			2: day-hospital
	Admission regime	Character (1)	3: home treatment
			4: day-surgery with overnight stay
**Drug prescriptions (*Assistenza Farmaceutica Territoriale and Farmaci a Erogazione Diretta*)**			
data_erog	Drug dispensing date	Date (10)	dd/mm/yyyy
atc	Anatomical Therapeutic Chemical classification	Character (7)	Alphanumeric
aic	Drug code (Autorizzazione all’Immissione in Commercio)	Character (9)	Alphanumeric
days	Prescription coverage days	Numeric	>0
pezzi	Number of packages	Numeric	>0
**Exemption for Pathology (*Esenzioni per Patologia*)**			
cod_esen	Exemption code	Character (8)	Alphanumeric
data_inizio	Start date	Date (10)	dd/mm/yyyy
data_fine	End date	Date (10)	dd/mm/yyyy
**Demographic population registry (*Anagrafe Sanitaria Assistiti*) and mortality registry**			
sesso	Sex	Character (1)	1: male, 2: female
datanas	Date of birth	Date (10)	dd/mm/yyyy
dinizio_resid	Residence start date	Date (10)	dd/mm/yyyy
dfine_resid	Residence end date	Date (10)	dd/mm/yyyy
reg_res	Region of residence	Character (3)	ISTAT code
com_res	Municipality of residence	Character (6)	ISTAT code
data_dec	Date of death	Date (10)	dd/mm/yyyy

**Table 3 jcm-14-06622-t003:** Algorithm definition for dementia, depression, diabetes and hypertension running on health administrative databases (drug prescription, hospital discharge and exemption for pathology). * Indicates that the ICD-9 codes with the same first digit are included.

Pathologies	Drug Prescription (ATC Code)	Hospital Discharge in Primary or Secondary Diagnosis (ICD-9 Code)	Exemption for Pathology
**Dementia**	At least two prescriptions in one year: N06DA04; N06DA03; N06DA02; N06DX0	290 *; 291.2; 294.0–294.21; 292.82; 331.0–331.2; 331.5, 331.7; 331.8; 331.82–331.9; 046.1	011; 029
**Depression**	At least 180 days of antidepressants prescriptions in one year: N06AA; N06AB; N06AX	296.20–296.25; 296.30–296.35; 300.4; 311; 296.6; 296.82; 296.90; 309.0; 309.1; 309.28	
**Diabetes**	At least two prescriptions in one year of all antidiabetic drugs: A10A; A10B	250 *	013
**Hypertension**	At least two prescriptions in one year: C02; C07; C08C; C09	401 *; 402 *; 403 *; 404 *; 405 *, 36211 Exclusion criteria (at least one episode of heart failure): 428 *, 36211; 39891; 40201; 40211; 40291; 40401; 40403; 40411; 40413; 40491; 40493	031

## Data Availability

The scripts developed for the disease classification algorithms have been made publicly available through the open-access repository Zenodo (https://zenodo.org/records/16036979, accessed on 10 September 2025).

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
