# Peer review of "Hypertension, Diabetes and Depression as Modifiable Risk Factors for Dementia: A Common Data Model Approach in a Population-Based Cohort, with Study Protocol and Preliminary Results"

_jcm, 2025, doi:10.3390/jcm14186622_

Round 1
Reviewer 1 Report
Comments and Suggestions for Authors
This manuscript presents a study protocol and preliminary results for a large-scale, population-based cohort study in Italy investigating the impact of hypertension, diabetes, depression, and their interactions on the risk of dementia. The study uses a Common Data Model (CDM) approach across five Italian regions/cities, leveraging Healthcare Utilization Databases (HUDs) to harmonize diverse administrative data sources. The authors describe in detail their CDM-based methodology, validated algorithms for disease classification, and statistical approaches including competing risks regression and meta-analysis. Preliminary descriptive data from over 3 million subjects are reported. The inclusion of over 3 million individuals across multiple Italian regions provides strong statistical power and representativity for dementia risk factor studies. Adopting a CDM across heterogeneous regional administrative databases is a robust and innovative methodological strength, enhancing data harmonization, reproducibility, and transparency. The plan to meta-analyze regional results allows for addressing heterogeneity and strengthens generalizability of findings.
Hereby are minor issues needed to be amended towards improving the manuscript’s quality:
- While the depression algorithm is pragmatic, the lack of clinical validation limits interpretability. Consider discussing potential misclassification bias this may introduce and its likely impact on risk estimates.
- The adjustment strategy includes age, sex, geography, and Charlson Comorbidity Index. However, key lifestyle confounders (e.g., smoking, education, physical activity) are not available in administrative data. The authors should acknowledge this limitation more explicitly and discuss potential residual confounding.
- The proposed 3-year lag sensitivity analysis is appropriate. The manuscript could expand discussing the biological plausibility and previous literature supporting this lag choice.
- While the waiver of informed consent is justified, a brief and short description of data security, anonymization protocols, and how GDPR compliance is ensured would strengthen ethical transparency.
- Define all abbreviations upon first use (e.g., PAF, PIF) in the abstract for clarity.
- In the Methods, specify the statistical software versions used.
- As describing the association between hypertension and dementia, you could mention in your discussion section the role of behavioral cardiology in the treatment strategy of these patients: please read and refer this recently published paper: https://www.mdpi.com/2075-4426/15/8/355 (https://doi.org/10.3390/jpm15080355)
Author Response
This manuscript presents a study protocol and preliminary results for a large-scale, population-based cohort study in Italy investigating the impact of hypertension, diabetes, depression, and their interactions on the risk of dementia. The study uses a Common Data Model (CDM) approach across five Italian regions/cities, leveraging Healthcare Utilization Databases (HUDs) to harmonize diverse administrative data sources. The authors describe in detail their CDM-based methodology, validated algorithms for disease classification, and statistical approaches including competing risks regression and meta-analysis. Preliminary descriptive data from over 3 million subjects are reported. The inclusion of over 3 million individuals across multiple Italian regions provides strong statistical power and representativity for dementia risk factor studies. Adopting a CDM across heterogeneous regional administrative databases is a robust and innovative methodological strength, enhancing data harmonization, reproducibility, and transparency. The plan to meta-analyze regional results allows for addressing heterogeneity and strengthens generalizability of findings.
Hereby are minor issues needed to be amended towards improving the manuscript’s quality:
- While the depression algorithm is pragmatic, the lack of clinical validation limits interpretability. Consider discussing potential misclassification bias this may introduce and its likely impact on risk estimates.
Thank you for the reviewer comment. We agree that the absence of clinical validation may introduce potential misclassification bias. We added this limitation in the Discussion section (pages 13-14, lines 365-374):
“Specifically, misclassification of depression may have occurred because antidepressant use was determined based on prescription records, which do not confirm actual medication intake or adherence to prescribed dosages. However, we defined depression exposure as having at least six months of continuous prescriptions, making it unlikely that individuals with such extended prescriptions did not take the medication. Additionally, while antidepressants can be prescribed for conditions other than depression, these alternative indications are generally treated with shorter courses, reducing the likelihood that long-term prescriptions represent non-depressive conditions. Similarly, for other conditions where validated algorithms were applied, some degree of misclassification may have occurred. To minimize this limitation, we selected algorithms with high specificity. Such potential misclassification is likely non-differential, which would tend to bias risk estimates toward the null. We will further assess the impact of these potential misclassifications during the analysis of association estimates.”
- The adjustment strategy includes age, sex, geography, and Charlson Comorbidity Index. However, key lifestyle confounders (e.g., smoking, education, physical activity) are not available in administrative data. The authors should acknowledge this limitation more explicitly and discuss potential residual confounding.
We are agree with the reviewer comment. We added the residual confounding limitation in the Discussion section (page 14, lines 377-380):
“Nonetheless, information on other potential confounders, such as lifestyle factors (e.g. smoking and physical activity), socioeconomic status, and education [Livistong 2024] was not available in the administrative data; therefore, our estimates of risk factor effects on the outcome may be subject to residual confounding.”
- The proposed 3-year lag sensitivity analysis is appropriate. The manuscript could expand discussing the biological plausibility and previous literature supporting this lag choice.
We agree that the discussion on the biological plausibility and supporting literature for the 3-year lag will strengthen the manuscript. We added the following text to the Discussion section (page 12, lines 308-312):
“The 3-year lag was selected based on the clinical history of dementia, recognizing that the prodromal phase, which includes subtle cognitive decline and other early symptoms, typically spans several years before a formal diagnosis is established [Duboisi2016]. A systematic review and meta-analysis [Kusoro2025] reported an average time to diagnosis of 3.5 years (95% Confidence Interval: 2.7–4.3 years) from symptom onset. This interval reflects the delay between initial symptom manifestation and clinical recognition.”
- While the waiver of informed consent is justified, a brief and short description of data security, anonymization protocols, and how GDPR compliance is ensured would strengthen ethical transparency.
We integrated the Method section (page 5, lines 167-172):
“All data were derived from HUDs managed within the INHS. To ensure compliance with the European Union General Data Protection Regulation (GDPR), Italian privacy laws, and the specific provisions outlined in Garante Privacy, Provvedimento n. 298/2024 (https://www.garanteprivacy.it/home/docweb/-/docweb-display/docweb/10016146), all personal identifiers were removed or pseudonymized prior to data analysis, preventing individual subject identification. Data access was strictly limited to authorized research personnel, and data processing adhered to principles of confidentiality, data minimization and security (GPDR-compliant servers).”
- Define all abbreviations upon first use (e.g., PAF, PIF) in the abstract for clarity.
As suggested we added the abbreviations in the Abstract: Italian National Health Service (INHS), Common Data Model (CDM) and Healthcare Utilization Databases (HUDs).
- In the Methods, specify the statistical software versions used.
We added the statistical software versions in the Statistical Analysis section (page 7, line 211):
“Statistical analyses will be performed using Stata SE 14.2 and SAS 9.4 service pack 8 software”.
- As describing the association between hypertension and dementia, you could mention in your discussion section the role of behavioral cardiology in the treatment strategy of these patients: please read and refer this recently published paper: https://www.mdpi.com/2075-4426/15/8/355 (https://doi.org/10.3390/jpm15080355)
We appreciate the reviewer’s suggestion to include a discussion on the role of behavioral cardiology in the treatment of these patients. We will consider the recent work by Fragoulis et al. (2025) in our forthcoming manuscript, which is currently in the analysis phase, when discussing the effect estimates between risk factors and dementia.
Reviewer 2 Report
Comments and Suggestions for Authors
Comments
The paper is scholarly and makes an essential contribution to dementia research. Its goal is to evaluate the influence of hypertension, diabetes, depression, and their interactions on the development of dementia in the Italian cohort study population using the CDM approach.
Abstract
- Authors can add more statistical findings in their research in the results section of the abstract.
Introduction
- In the first paragraph, introduce the pathology of dementia and its types.
Materials and Methods
- 2.7 and 2.8 subheadings can be moved to the end of the manuscript
Results
- In Table 1, sex already includes females, not males, in the cohort study?
- The results were documented, but you mention that a regression model was used with covariate adjustments. However, I do not see the results for the regression model showing how the models performed based on the covariate adjustments, as mentioned in lines 191-194. Please clarify this.
Discussion
- It is well written and organized.
Conclusion
- No comments for this section, well written and clear.
Author Response
The paper is scholarly and makes an essential contribution to dementia research. Its goal is to evaluate the influence of hypertension, diabetes, depression, and their interactions on the development of dementia in the Italian cohort study population using the CDM approach.
Thank you for the reviewer’s encouraging feedback.
Abstract
- Authors can add more statistical findings in their research in the results section of the abstract.
As suggested, we added in the Abstract more results (page 2, lines 58-59):
“On January 1st, 2011, the prevalence of individuals aged ≥ 50 years with dementia ranged from 8.7 to 14.7 per 1,000 population.”
Introduction
- In the first paragraph, introduce the pathology of dementia and its types.
As suggested, we added in the Introduction section the pathology of dementia and its types (page 3, lines 82-86):
“It results from diverse pathological changes in the brain, including the accumulation of abnormal proteins, neuronal loss, and vascular damage. The most common types of dementia include Alzheimer’s disease, characterized by amyloid-beta plaques and neurofibrillary tau tangles; vascular dementia, caused by cerebrovascular disease; Lewy body dementia, marked by alpha-synuclein-containing Lewy bodies; and frontotemporal dementia, involving degeneration of the frontal and temporal lobes. (Scheltens et al., 2021).”
Materials and Methods
- 2.7 and 2.8 subheadings can be moved to the end of the manuscript
We moved 2.7 and 2.8 subheadings to the end of the manuscript.
Results
- In Table 1, sex already includes females, not males, in the cohort study?
In Table 1, we reported the total population along with the number and percentage of females. We did not include data on males to avoid redundancy, as these values can be easily inferred.
- The results were documented, but you mention that a regression model was used with covariate adjustments. However, I do not see the results for the regression model showing how the models performed based on the covariate adjustments, as mentioned in lines 191-194. Please clarify this.
This paper presents the study protocol along with some preliminary descriptive results. The regression model analyses are currently underway and will be reported in a future publication.
Discussion
- It is well written and organized.
Conclusion
- No comments for this section, well written and clear.
Round 2
Reviewer 1 Report
Comments and Suggestions for Authors
All raised issues have been amended. Congratulations for your work.
Reviewer 2 Report
Comments and Suggestions for Authors
The authors addressed all the comments. I don't have any more comments.